# Adan: Adaptive Nesterov Momentum Algorithm for Faster Optimizing Deep Models

**Xingyu Xie**[1,2*]    **Pan Zhou**[1*]    **Huan Li**[3]    **Zhouchen Lin**[2⋆]    **Shuicheng Yan**[1⋆]
[1]Sea AI Lab    [2]Peking University    [3]Nankai University
{xyxie,zhoupan,yansc}@sea.com    {xyxie,zlin}@pku.cn    lihuanss@nankai.edu.cn

## Abstract

Adaptive gradient algorithms [1–4] combine the moving average idea with heavy ball acceleration to estimate accurate first- and second-order moments of gradient for accelerating convergence. But Nesterov acceleration which converges faster than heavy ball acceleration in theory [5] and also in many empirical cases [6] is much less investigated under the adaptive gradient setting. In this work, we propose the ADAptive Nesterov momentum algorithm (Adan) to speed up the training of deep neural networks. Adan first reformulates the vanilla Nesterov acceleration to develop a new Nesterov momentum estimation (NME) method that avoids the extra computation and memory overhead of computing gradient at the extrapolation point. Then Adan adopts NME to estimate the first- and second-order gradient moments in adaptive gradient algorithms for convergence acceleration. Besides, we prove that Adan finds an $\epsilon$-approximate stationary point within $\mathcal{O}(\epsilon^{-4})$ stochastic gradient complexity on the non-convex stochastic problems, matching the best-known lower bound. Extensive experimental results show that Adan surpasses the corresponding SoTA optimizers for vision, language, and RL tasks and sets new SoTAs for many popular networks and frameworks, *e.g.* ResNet [7], ConvNext [8], ViT [9], Swin [10], MAE [11], Transformer-XL [12] and BERT [13]. More surprisingly, Adan can use half of the training cost (epochs) of SoTA optimizers to achieve higher or comparable performance on ViT, ResNet, MAE, *etc*, and also shows great tolerance to a large range of minibatch size, *e.g.* from 1k to 32k. Code is released at https://github.com/sail-sg/Adan.

## 1 Introduction

Deep neural networks (DNNs) have made remarkable success in many fields, *e.g.* computer vision [7, 8, 14–16] and natural language processing [17, 18]. A noticeable part of such success is contributed by the stochastic gradient based optimizers which find satisfactory solutions with high efficiency. Starting from AdaGrad [19] and RMSProp [20], adaptive gradient algorithms [1–3, 19–25] have gained wide attention with faster convergence speed. They adjust the learning rate for each gradient coordinate according to the current geometry curvature of the loss objective. Indeed, Adam [1] and AdamW [3], as two representatives which often offer fast convergence speed across many DNN frameworks, have become the default choice to train CNNs and ViTs [9], respectively.

However, none of the above optimizers can always stay undefeated among all its competitors across different network architectures and application settings. For instance, for vanilla ResNet, SGD often achieves better generalization performance than adaptive gradient algorithms such as Adam, whereas on vision transformers (ViTs) [9, 10, 26], SGD often fails and AdamW is the dominant optimizer with higher and more stable performance. Moreover, these commonly used optimizers usually fail for

---

⋆Equal contribution. Xingyu did this work during an internship at Sea AI Lab.
⋆Co-corresponding authors.

Has it Trained Yet? Workshop at the Conference on Neural Information Processing Systems (NeurIPS 2022).

large-batch training, but which is a default setting of the prevalent distributed training. Although there is some performance degradation, we still tend to choose the large-batch setting for large-scale deep learning training tasks due to the unaffordable training time. Though some methods, *e.g.* LARS [27] and LAMB [4], have been proposed to handle large batch sizes, their performance often varies significantly across batch sizes. This performance inconsistency increases the training cost and engineering burden, since one usually has to try various optimizers for different architectures.

When we rethink the current adaptive gradient algorithms, we find that they mainly combine the moving average idea with the heavy ball acceleration technique to estimate the first- and second-order moments of the gradient [1–4]. However, previous studies [5, 28, 29] have revealed that Nesterov acceleration can theoretically achieve a faster convergence speed than heavy ball acceleration, as it uses gradient at an extrapolation point of the current solution and sees a slight "future". Moreover, a recent work [6] has shown the potential of Nesterov acceleration for large-batch training [30]. Thus we are inspired to consider efficiently integrating Nesterov acceleration with adaptive algorithms.

**Contributions: 1)** We propose an efficient dnn optimizer, named Adan, to train DNNs. Adan develops a Nesterov momentum estimation method to estimate stable and accurate first- and second-order gradient moments in adaptive algorithms for acceleration. **2)** Adan enjoys provably faster convergence speed than previous adaptive algorithms, *e.g.* Adam. **3)** Empirically, Adan shows superior performance over the SoTA deep optimizers across vision, language, and RL tasks.

## 2 Methodology

In this work, we study the following regularized nonconvex optimization problem:

$$\min_{\boldsymbol{\theta}} F(\boldsymbol{\theta}) \coloneqq \mathbb{E}_{\boldsymbol{\zeta} \sim \mathcal{D}} \left[ f(\boldsymbol{\theta}, \boldsymbol{\zeta}) \right] + \frac{\lambda}{2} \|\boldsymbol{\theta}\|^2, \tag{1}$$

where loss $f(\cdot, \cdot)$ is differentiable and possibly nonconvex, data $\boldsymbol{\zeta}$ is drawn from an unknown distribution $\mathcal{D}$, $\boldsymbol{\theta}$ is learnable parameters, and $\|\cdot\|$ is the classical $\ell_2$ norm. Here we consider the square $\ell_2$ regularizer as it can improve generalization performance and is widely used in practice [3].

### 2.1 Preliminaries

Taking a deeper step into Adam, one can easily observe that the key moving average idea in Adam is similar to the classical (stochastic) heavy-ball acceleration (HBA) technique [31]:

$$\text{HBA:} \quad \mathbf{g}_k = \nabla f(\boldsymbol{\theta}_k) + \boldsymbol{\xi}_k, \qquad \mathbf{m}_k = (1 - \beta_1)\mathbf{m}_{k-1} + \mathbf{g}_k, \qquad \boldsymbol{\theta}_{k+1} = \boldsymbol{\theta}_k - \eta\mathbf{m}_k,$$

where $\mathbf{g}_k$ is the minibatch gradient $\mathbf{g}_k \coloneqq \mathbb{E}_{\boldsymbol{\zeta} \sim \mathcal{D}}[\nabla f(\boldsymbol{\theta}_k, \boldsymbol{\zeta})] + \boldsymbol{\xi}_k$, $\boldsymbol{\xi}_k$ is the gradient noise, and the scalar constant $\eta$ is the base learning rate.

In addition to HBA, Nesterov's accelerated (stochastic) gradient descent (AGD) [5, 28, 29] is another popular acceleration technique in the optimization community:

$$\text{AGD:} \ \mathbf{g}_k = \nabla f(\boldsymbol{\theta}_k - \eta(1 - \beta_1)\mathbf{m}_{k-1}) + \boldsymbol{\xi}_k, \ \mathbf{m}_k = (1 - \beta_1)\mathbf{m}_{k-1} + \mathbf{g}_k, \ \boldsymbol{\theta}_{k+1} = \boldsymbol{\theta}_k - \eta\mathbf{m}_k. \tag{2}$$

Unlike HBA, AGD uses the gradient at the extrapolation point $\boldsymbol{\theta}'_k = \boldsymbol{\theta}_k - (1 - \beta_1)(\boldsymbol{\theta}_k - \boldsymbol{\theta}_{k-1})$. Hence AGD sees a slight "future" to converge faster. Indeed, AGD theoretically converges faster than HBA and achieves optimal convergence rate on the general smooth convex problems [5]. Meanwhile, since the over-parameterized DNNs have been observed/proved to have many convex-alike local basins [32–39], AGD seems more suitable than HBA for DNNs.

For large-batch training, the recent work [6] shows that AGD has the potential to achieve comparable performance to some specifically designed optimizers, *e.g.* LARS and LAMB. With its advantage in convergence and large-batch training, we consider applying AGD to improve adaptive algorithms.

### 2.2 Adaptive Nesterov Momentum Algorithm

**Main iteration.** We temporarily set $\lambda = 0$ in Eqn. (1). As aforementioned, AGD computes gradient at an extrapolation point $\boldsymbol{\theta}'_k$ instead of the current iterate $\boldsymbol{\theta}_k$, which however brings extra computation and memory overhead for computing $\boldsymbol{\theta}'_k$ and preserving both $\boldsymbol{\theta}_k$ and $\boldsymbol{\theta}'_k$. To solve the issue, we reformulate AGD (2) into its equivalent (See Lemma 1 in Appendix) but more DNN-efficient version:

Reformulated AGD: $\mathbf{m}_k = (1 - \beta_1)\mathbf{m}_{k-1} + [\mathbf{g}_k + (1 - \beta_1)(\mathbf{g}_k - \mathbf{g}_{k-1})], \quad \boldsymbol{\theta}_{k+1} = \boldsymbol{\theta}_k - \eta\mathbf{m}_k.$

**Algorithm 1: Adan** (Adaptive Nesterov Momentum Algorithm)

---

**Input:** initialization $\boldsymbol{\theta}_0$, step size $\eta$, momentum $(\beta_1, \beta_2, \beta_3) \in [0,1]^3$, weight decay $\lambda_k > 0$.

**Output:** some average of $\{\boldsymbol{\theta}_k\}_{k=1}^K$.

1  **while** $k < K$ **do**
2  $\quad$ compute the stochastic gradient estimator $\mathbf{g}_k$ at $\boldsymbol{\theta}_k$;
3  $\quad$ $\mathbf{m}_k = (1 - \beta_1)\mathbf{m}_{k-1} + \beta_1 \mathbf{g}_k$ $\qquad\qquad$ /* set $\mathbf{m}_0 = \mathbf{g}_0$ */;
4  $\quad$ $\mathbf{v}_k = (1 - \beta_2)\mathbf{v}_{k-1} + \beta_2(\mathbf{g}_k - \mathbf{g}_{k-1})$ $\qquad$ /* set $\mathbf{v}_1 = \mathbf{g}_1 - \mathbf{g}_0$ */;
5  $\quad$ $\mathbf{n}_k = (1 - \beta_3)\mathbf{n}_{k-1} + \beta_3 [\mathbf{g}_k + (1 - \beta_2)(\mathbf{g}_k - \mathbf{g}_{k-1})]^2$ $\qquad$ /* set $\mathbf{n}_0 = \mathbf{g}_0^2$ */;
6  $\quad$ $\boldsymbol{\eta}_k = \eta / (\sqrt{\mathbf{n}_k} + \varepsilon)$ $\qquad$ /* $\varepsilon > 0$ is for stabilize training */;
7  $\quad$ $\boldsymbol{\theta}_{k+1} = (1 + \lambda_k \eta)^{-1} [\boldsymbol{\theta}_k - \boldsymbol{\eta}_k \circ (\mathbf{m}_k + (1 - \beta_2)\mathbf{v}_k)]$;
8  **end while**

---

where $\mathbf{g}_k = \mathbb{E}_{\boldsymbol{\zeta} \sim \mathcal{D}}[\nabla f(\boldsymbol{\theta}_k, \boldsymbol{\zeta})] + \boldsymbol{\xi}_k$. The main idea here is that we maintain $(\boldsymbol{\theta}_k - \eta(1 - \beta_1)\mathbf{m}_{k-1})$ rather than $\boldsymbol{\theta}_k$ in vanilla AGD per iteration, as there is no difference between them when the algorithm converges. Like Adam, by regarding $\mathbf{g}'_k = \mathbf{g}_k + (1 - \beta_1)(\mathbf{g}_k - \mathbf{g}_{k-1})$ as the current stochastic gradient and movingly averaging $\mathbf{g}'_k$ to estimate the first- and second-moments of gradient, we obtain

$$\text{Vanilla Adan:} \begin{cases} \mathbf{m}_k = (1 - \beta_1)\mathbf{m}_{k-1} + \beta_1 [\mathbf{g}_k + (1 - \beta_1)(\mathbf{g}_k - \mathbf{g}_{k-1})] \\ \mathbf{n}_k = (1 - \beta_3)\mathbf{n}_{k-1} + \beta_3 (\mathbf{g}_k + (1 - \beta_1)[\mathbf{g}_k - \mathbf{g}_{k-1}])^2 \\ \boldsymbol{\eta}_k = \eta / (\sqrt{\mathbf{n}_k} + \varepsilon), \quad \boldsymbol{\theta}_{k+1} = \boldsymbol{\theta}_k - \boldsymbol{\eta}_k \circ \mathbf{m}_k. \end{cases}$$

The main difference of Adan with Adam-type methods is that as compared in Eqn. (3), the moment $\mathbf{m}_k$ of Adan is the average of $\{\mathbf{g}_t + (1 - \beta_1)(\mathbf{g}_t - \mathbf{g}_{t-1})\}_{t=1}^k$ while those of Adam-type are the average of $\{\mathbf{g}_t\}_{t=1}^k$. So is their second-order term $\mathbf{n}_k$.

$$\mathbf{m}_k = \begin{cases} \sum_{t=0}^k c_{k,t}[\mathbf{g}_t + (1 - \beta_1)(\mathbf{g}_t - \mathbf{g}_{t-1})], & \text{Adan,} \\ \sum_{t=0}^k c_{k,t}\mathbf{g}_t, & \text{Adam,} \end{cases} \quad c_{k,t} = \begin{cases} \beta_1 (1 - \beta_1)^{(k-t)} & t > 0, \\ (1 - \beta_1)^k & t = 0, \end{cases} \quad (3)$$

The first-order moment $\mathbf{m}_k = \sum_{t=0}^k c_{k,t}[\mathbf{g}_t + (1 - \beta_1)(\mathbf{g}_t - \mathbf{g}_{t-1})]$ consists of two terms, i.e., gradient term $\mathbf{g}_t$ and gradient difference term $(\mathbf{g}_t - \mathbf{g}_{t-1})$, which actually have different physic meanings. So we further decouple them for greater flexibility and also better trade-off between them:

$$(\boldsymbol{\theta}_{k+1} - \boldsymbol{\theta}_k)/\boldsymbol{\eta}_k = \sum_{t=0}^k \left[ c_{k,t}\mathbf{g}_t + (1 - \beta_2)c'_{k,t}(\mathbf{g}_t - \mathbf{g}_{t-1}) \right] = \mathbf{m}_k + (1 - \beta_2)\mathbf{v}_k,$$

where $c'_{k,t} = \beta_2 (1 - \beta_2)^{(k-t)}$ for $t > 0$, $c'_{k,t} = (1 - \beta_2)^k$ for $t = 0$, and $\mathbf{m}_k$ and $\mathbf{v}_k$ are defined as

$$\mathbf{m}_k = (1 - \beta_1)\mathbf{m}_{k-1} + \beta_1 \mathbf{g}_k, \qquad \mathbf{v}_k = (1 - \beta_2)\mathbf{v}_{k-1} + \beta_2(\mathbf{g}_k - \mathbf{g}_{k-1}).$$

This change for a flexible estimation does not impair convergence speed. As we show in Theorem 1 (see Sec. C in Appendix), the complexity of Adan under this change matches the lower complexity bound. We do not separate the gradients and their difference in the second-order moment $\mathbf{n}_k$, since $\mathbb{E}(\mathbf{n}_k)$ contains the correlation term $\text{Cov}(\mathbf{g}_k, \mathbf{g}_{k-1}) \neq 0$ which may have statistical significance.

**Decay Weight by Proximation.** As observed in AdamW, decoupling the optimization objective and simple-type regularization (*e.g.* $\ell_2$ regularizer) can largely improve the generalization performance. Here we follow this idea but from a rigorous optimization perspective. Intuitively, at each iteration, we minimize the first-order approximation of $F(\cdot)$ at the point $\boldsymbol{\theta}_k$:

$$\boldsymbol{\theta}_{k+1} = \boldsymbol{\theta}_k - \boldsymbol{\eta}_k \circ \bar{\mathbf{m}}_k = \arg\min_{\boldsymbol{\theta}} \left( F(\boldsymbol{\theta}_k) + \langle \bar{\mathbf{m}}_k, \boldsymbol{\theta} - \boldsymbol{\theta}_k \rangle + \frac{1}{2\eta} \|\boldsymbol{\theta} - \boldsymbol{\theta}_k\|_{\sqrt{\mathbf{n}_k}}^2 \right),$$

where $\|\mathbf{x}\|_{\sqrt{\mathbf{n}_k}}^2 := \langle \mathbf{x}, \sqrt{\mathbf{n}_k + \varepsilon} \circ \mathbf{x} \rangle$ and $\bar{\mathbf{m}}_k := \mathbf{m}_k + (1 - \beta_2)\mathbf{v}_k$ is the first-order derivative of $F(\cdot)$ in some sense. Follow the idea of proximal gradient descent [40, 41], we decouple the $\ell_2$ regularizer from $F(\cdot)$ and only linearize the loss function $f(\cdot)$:

$$\boldsymbol{\theta}_{k+1} = \arg\min_{\boldsymbol{\theta}} \left( \frac{\lambda_k}{2} \|\boldsymbol{\theta}\|_{\sqrt{\mathbf{n}_k}}^2 + \langle \bar{\mathbf{m}}_k, \boldsymbol{\theta} - \boldsymbol{\theta}_k \rangle + \frac{1}{2\eta} \|\boldsymbol{\theta} - \boldsymbol{\theta}_k\|_{\sqrt{\mathbf{n}_k}}^2 \right) = \frac{\boldsymbol{\theta}_k - \boldsymbol{\eta}_k \circ \bar{\mathbf{m}}_k}{1 + \lambda_k \eta}, \quad (4)$$

where $\lambda_k > 0$ is the weight decay constant at the $k$-th iteration. One can find that the optimization objective of Separated Regularization at the $k$-th iteration is changed from the vanilla "static" function

Table 1: Top-1 Acc. (%) of ResNet and ConvNext on ImageNet. ∗ and ⋄ are reported in [42], [8].

| Epoch | ResNet-50 | | | ResNet-101 | | |
|---|---|---|---|---|---|---|
| | 100 | 200 | 300 | 100 | 200 | 300 |
| SAM | 77.3 | 78.7 | 79.4 | 79.5 | 81.1 | 81.6 |
| SGD-M | 77.0 | 78.6 | 79.3 | 79.3 | 81.0 | 81.4 |
| Adam | 76.9 | 78.4 | 78.8 | 78.4 | 80.2 | 80.6 |
| AdamW | 77.0 | 78.9 | 79.3 | 78.9 | 79.9 | 80.4 |
| LAMB | 77.0 | 79.2 | 79.8* | 79.4 | 81.1 | 81.3* |
| **Adan (ours)** | **78.1** | **79.7** | **80.2** | **79.9** | **81.6** | **81.8** |

| Epoch | ConvNext Tiny | |
|---|---|---|
| | 150 | 300 |
| AdamW [3, 8] | 81.2 | 82.1⋄ |
| **Adan (ours)** | **81.7** | **82.4** |

| Epoch | ConvNext Small | |
|---|---|---|
| | 150 | 300 |
| AdamW [3, 8] | 82.2 | 83.1⋄ |
| **Adan (ours)** | **82.5** | **83.3** |

Table 2: Top-1 Acc. (%) of ViT and Swin on ImageNet. ∗ and ⋄ are respectively reported in [9], [10].

| Epoch | ViT Small | | ViT Base | | Swin Tiny | | Swin small | | Swin Base | |
|---|---|---|---|---|---|---|---|---|---|---|
| | 150 | 300 | 150 | 300 | 150 | 300 | 150 | 300 | 150 | 300 |
| AdamW [3, 9, 10] | 78.3 | 79.9* | 79.5 | 81.8* | 79.9 | 81.2⋄ | 82.1 | 83.2⋄ | 82.6 | 83.5⋄ |
| **Adan (ours)** | **79.6** | **80.9** | **81.7** | **82.3** | **81.3** | **81.6** | **82.9** | **83.7** | **83.3** | **83.8** |

$F(\cdot)$ in (1) to a "dynamic" function $F_k(\cdot)$, which adaptively regularizes the coordinates with larger gradient square terms more. We summarize our Adan in Algorithm 1.

**Convergence Analysis:** As shown in Theorems 1 in Appendix. C, the convergence speed of Adan matches the best-known theoretical lower bound for non-convex stochastic optimization problems. This conclusion is still valid when it also uses the decoupled weight decay.

## 3 Experimental Results

For all the tested vision tasks, NLP, and RL tasks, we only replace the default optimizer with our Adan, and do not make other changes, *e.g.* network architectures and data augmentation.

**Vision Results.** 1) supervised settings: we report the results on CNN-type architectures and ViTs in Tables 1 and 2, respectively. 2) self-supervised settings: we follow the MAE training framework to pretrain and fine-tune ViT-B and ViT-L, and report results in Table 3. All these results show that *in most cases, Adan can use half of the training cost (epochs) of SoTA optimizers to achieve higher or comparable performance on ViT, ResNet, MAE,* etc.

Table 3: Top-1 Acc. (%) of ViT-B and ViT-L trained by self-supervised MAE on ImageNet.

| Epoch | MAE-ViT-B | | | MAE-ViT-L | |
|---|---|---|---|---|---|
| | 300 | 800 | 1600 | 800 | 1600 |
| AdamW | 82.9 | — | 83.6 | 85.4 | 85.9 |
| **Adan** | **83.4** | **83.8** | — | **85.9** | — |

**NLP Results.** 1) supervised settings: we investigate the performance of Adan on Transformer-XL, and report the results in Table 4. 2) self-supervised settings: we use Adan to train BERT from scratch, and report the results in Table 5. *For all NLP tasks, Adan achieves higher performance than the default SoTAs, and suppress Adam within half training steps on Transformer-XL.*

Table 4: Test PPL for Transformer-XL-base model on WikiText-103.

| Transformer-XL | Training Steps | | |
|---|---|---|---|
| | 50k | 100k | 200k |
| Adam [1] | 28.5 | 25.5 | 24.2 |
| **Adan (ours)** | **26.2** | **24.2** | **23.5** |

Table 5: Results (the higher, the better) of BERT-base model on the development set of GLUE.

| BERT-base | MNLI | QNLI | QQP | RTE | SST-2 | CoLA | STS-B | **Average** |
|---|---|---|---|---|---|---|---|---|
| Adam [1] (from [43]) | 83.7/84.8 | 89.3 | 90.8 | 71.4 | 91.7 | 48.9 | 91.3 | 81.5 |
| Adam [1] (reproduced) | 84.9/84.9 | 90.8 | 90.9 | 69.3 | 92.6 | 58.5 | 88.7 | 82.5 |
| **Adan (ours)** | **85.7/85.6** | **91.3** | **91.2** | **73.3** | **93.2** | **64.6** | **89.3** | **84.3 (+1.8)** |

**RL Results.** We replace the default Adam optimizer in PPO [44] (one of the most popular policy gradient method), and do not make other change in PPO. Fig. 1 shows that *on representative MuJoCo games, PPO-Adan achieves much higher rewards than PPO with Adam as its optimizer.*

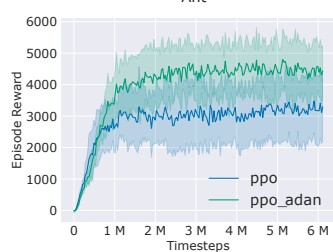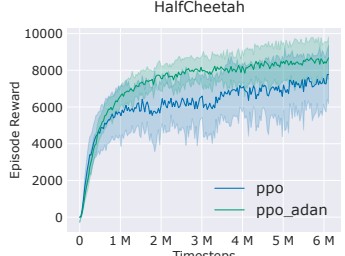

Figure 1: PPO with Adam and Adan as its optimizer.

**More Extra Results.** Due to space limitation, we defer more extra results on vision, NLP and RL tasks (*e.g.* results with large batch size, loss curve, ablation study, *etc.*) to Appendix.

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
