# OpenReview forum: "Adan: Adaptive Nesterov  Momentum Algorithm for  Faster Optimizing Deep Models"
_NeurIPS.cc/2022/Workshop/HITY — HITY Workshop NeurIPS 2022_

### Official Review · Reviewer_y9J3 · 2022-10-17
**Adaptive gradient method based on Nesterov acceleration**

**Rating:** 1
**Confidence:** 3

**Review:**

The authors present a new adaptive gradient method based on Nesterov acceleration. The results look very promising and the work is well presented.

---

### Official Review · Reviewer_98xo · 2022-10-19

**Rating:** 1
**Confidence:** 5

**Review:**

Overall looks fine to me. Would like to see more info on tuning and how the comparisons were done:

- "Moreover, these commonly used optimizers usually fail for large-batch training, but which is a default setting of the prevalent distributed training." what is the evidence for these optimizers failing at large batches? many LLMs nowadays train with 'large batches', and some works have shown that nesterov/adam work fine at 'large batches' on resnet/BERT https://arxiv.org/abs/2102.06356
- your claim "At present, LAMB is the most effective optimizer for large batch size.", is this specific to ViT? https://arxiv.org/abs/2102.06356 and other works seems to train models fine at batch sizes up to 32k (however, the meaning of a batch size is not useful across workloads). also, how were the LAMB results from table 7 tuned? did you retune adan or lamb at each batch size?
- "Adan enjoys provably faster convergence speed than previous adaptive algorithms" you make claims that the convergence speed is faster, but many of your results involve better final validation/test metric values, which implies better generalization instead of convergence. I am certain that your algorithm achieves both, but it could also be nice to include some results that are "time to target" where you fix a target test metric value and measure the wallclock time or number of steps to reach the value.
- you mention in the first paragraph of appendix D that you tune some hparams in adan. how did you tune them (random search, grid search, etc)? was the same tuning budget given to adan and the baseline being compared to?

---

### Decision · Program_Chairs · 2022-10-20

Accept